# The Complete Mitochondrial Genome of *Mytilisepta virgata* (Mollusca: Bivalvia), Novel Gene Rearrangements, and the Phylogenetic Relationships of Mytilidae

**DOI:** 10.3390/genes14040910

**Published:** 2023-04-13

**Authors:** Minhui Xu, Zhongqi Gu, Ji Huang, Baoying Guo, Lihua Jiang, Kaida Xu, Yingying Ye, Jiji Li

**Affiliations:** 1National Engineering Research Center for Marine Aquaculture, Zhejiang Ocean University, Zhoushan 316022, China; 2Shengsi Marine Science and Technology Institute, Shengsi, Zhoushan 202450, China; 3Key Laboratory of Sustainable Utilization of Technology Research for Fisheries Resources of Zhejiang Province, Scientific Observing and Experimental Station of Fishery Resources for Key Fishing Grounds, Ministry of Agriculture and Rural Affairs of China, Zhejiang Marine Fisheries Research Institute, Zhoushan 316021, China

**Keywords:** Mytilidae, *Mytilisepta virgate*, mitogenome, gene rearrangement, phylogenetic

## Abstract

The circular mitochondrial genome of *Mytilisepta virgata* spans 14,713 bp, which contains 13 protein-coding genes (PCGs), 2 ribosomal RNA genes, and 22 transfer RNA genes. Analysis of the 13 PCGs reveals that the mitochondrial gene arrangement of *Mytilisepta* is relatively conserved at the genus level. The location of the *atp8* gene in *Mytilisepta keenae* differs from that of other species. However, compared with the putative molluscan ancestral gene order, *M. virgata* exhibits a high level of rearrangement. We constructed phylogenetic trees based on concatenated 12 PCGs from Mytilidae. As a result, we found that *M. virgata* is in the same clade as other *Mytilisepta* spp. The result of estimated divergence times revealed that *M. virgata* and *M. keenae* diverged around the early Paleogene period, although the oldest *Mytilisepta* fossil was from the late or upper Eocene period. Our results provide robust statistical evidence for a sister-group relationship within Mytilida. The findings not only confirm previous results, but also provide valuable insights into the evolutionary history of Mytilidae.

## 1. Introduction

The mitochondrial genome (mitogenome) is regarded as a good model of phylogenetics in the investigation of species due to its small molecular weight and maternal inheritance [1]. With the rapid advancement of next-generation sequencing (NGS) technology, there was a significant reduction in the cost of DNA sequencing, and analyses of the complete mitochondrial genomes (mitogenomes) have gained popularity in recent years for phylogenetic investigations [2,3]. Compared to nuclear genes, mitochondrial DNA sequences can provide more informative data for phylogenetic analysis, as well as multiple structural genomic features [4,5], such as gene length, compositional features, gene order, and the secondary structure of the encoded RNA. The typical metazoan mitogenome is generally a circular, double-stranded molecule ranging from 15 to 20 kb in size, and contains 37 genes: 2 for rRNAs, 13 for proteins, and 22 for tRNAs [6]. The family *Mytilidae* constitutes a major clade within Mytilida and includes approximately 88 extant genera containing 1632 species (World Register of Marine Species, https://www.marinespecies.org/ (accessed 6 September 2022)). The species among *Mytilidae* are represented by one of the largest numbers of cultivated and marketed bivalves, such as *Mytilus* spp., which are widely distributed in cold and temperate waters throughout the world’s oceans [7]. Furthermore, species of *Perna* spp. are well researched because of their significant economic and social importance in the aquaculture and fishing sectors [8]. The dominant rocky shore bivalve *Xenostrobus securis* Lamarck, 1819, and the freshwater bivalve *Limnoperna fortunei* Dunker, 1857, became invasive in the Northern Hemisphere [9]. Non-obvious small taxa such as *Modiolarca subpicta*, Cantraine, 1835 [10], and the taxa *Mytilisepta virgata* Wiegmann, 1837, have attracted less scientific attention. Many bivalve species exhibit a distinctive pattern of mitochondrial inheritance known as doubly uniparental inheritance (DUI) [11]. At present, no studies have reported whether there is DUI in the mitochondria of *M. virgata*.

The genus *Mytilisepta* (Habe, 1951) belongs to the family Mytilidae, and it contains three species: *M. keenae* Nomura, 1936; *M. bifurcata* Conrad, 1837; and *M. virgata*. The finely ribbed black mussel *M. virgata*, also known as *Septifer virgatus*, is widely distributed in Japan, Korea, and several locations along the coast of China and Hong Kong [12,13,14,15]. It usually forms in large mussel beds that contribute to providing refuge and a suitable habitat for diverse invertebrate marine species [16]. In comparison to other lower tidal sympatric mussel species, this mussel species is flatter ventrally, has a wider and firmer shell, and has a stronger byssal attachment for improved physical stability to deal with its rough environment [17]. From a taxonomic perspective, *M. virgata* has long been identified as a species from *Septifer* and was placed within the subfamily *Septiferinae* [18]. However, *M. virgata* and the majority of the other species formerly placed within *Septiferinae* were then moved to *Mytilinae Rafinesque*, 1815. It is thought that *Mytilisepta* is a junior synonym of *Septifer*, according to the revised hierarchical classification of the NCBI Taxonomy. An anterior internal umbilical septum of each valve is the most distinctive feature of the shells of the *Septifer* and *Mytilisepta* species, which provides the subfamily with its name [19]. Until now, only linear mitochondrial sequences and individual genes were available for species of *M. virgata* in the NCBI database. 

In this study, we aimed to sequence the complete mitochondrial genome of *M. virgata* to increase taxon sampling for the genus *Mytilisepta*. We also analyzed the genomic features and evolutionary pattern of its mitogenome, including gene order, nucleotide composition, codon usage, and the secondary structure of tRNAs. Moreover, we performed a phylogenetic analysis of the subclass Pteriomorphia to evaluate the phylogenetic position of *M. virgata*. In addition, we integrated the gene arrangement of mitogenomes during evolution in Mytilidae to obtain accurate evolutionary relationships and determine the divergence time of the major lineages of *Mytilisepta*. Furthermore, we incorporated the new mitogenomes into the Mytilidae dataset to assist in clarifying contentious phylogenetic relationships associated with the family Mytilidae. In a broader sense, the phylogenetic tree of *M. virgata* and its corresponding gene order is important for higher taxonomic level genomic and systematic studies due to the uncertainties currently associated with deep relationships within the family Mytilidae.

## 2. Materials and Methods

### 2.1. Collection of Samples and DNA Extraction

Samples of *M. virgata* were collected from Gouqi Island (N 30°43′1.64″, E 122°46′3.25″) in the Zhejiang province of China in November 2020. The specimens were stored in 95% ethanol before DNA extraction. The total genomic DNA was extracted from the adductor muscle using the rapid salting-out method [20]. Then, the quality of the DNA was checked with 1% agarose gel electrophoresis and the DNA was preserved at −20 °C.

### 2.2. Sequencing, Assembly, and Annotation of Mitochondrial Genomes

The complete mitogenome of *M. virgata* was sequenced using next-generation sequencing by Origingene Bio-pharm Technology Co., Ltd. (Shanghai, China). The 1 μg DNA was first cut into approximately 300–500 bp using a physical ultrasonic method (Covaris M220). Then, the TruSeq Nano DNA Sample Prep Kit was used to complete the leveling of the 3′ end with a nucleotide before the ligation of index connectors. The DNA fragments after ligation were amplified by PCR (eight cycles). Subsequently, quantitative analysis was performed using TBS380 (Picogreen) and bridge PCR amplification on a cBot solid phase vector to generate clusters. Finally, 2 × 150 bp sequencing was conducted on the Illumina NovaSeq 6000 platform using total genomic DNA. Data quality control was performed using Trimmomatic v0.39 [21], which filtered out the adapter sequence in reads; the quality value of sequencing reads was less than that of the terminal reads of Q20 and the sequencing linker sequences. The clean data were reassembled using NOVOPlasty software (https://github.com/ndierckx/NOVOPlasty (accessed 6 September 2022)) without sequencing adapters [22]. To ensure the correctness of the sequence, we conducted NCBI BLAST searches based on the *cox1* barcode sequence to identify the mitogenome sequence. The MITOS web server (http://mitos2.bioinf.uni-leipzig.de/index.py (accessed 6 September 2022)) was utilized to predict the new mitogenome of protein-coding genes (PCGs), tRNA, and rRNA genes, and redundancy for the predicted initial genes was removed. The initial and terminal codon positions were manually corrected by comparing them with sequences from other mussels to obtain a highly accurate set of conserved genes [23]. Moreover, we manually annotated sequences lacking *atp8* by scanning the intergenic regions. ORFfinder (https://www.ncbi.nlm.nih.gov/orffinder/ (accessed 15 March 2023)) was used to find the ORFs (open reading frames). The starting codon of the *atp8* sequences was corrected according to the sequences of related species. 

### 2.3. Visualization and Comparative Analysis of the Genome 

The sequence features of the mitochondrial circular genome of *M. virgata* were shown using the online CGView server [24]. The relative synonymous codon usage (RSCU) values were analyzed using MEGA X [25]. The composition skew values were calculated based on the ratio of AT-skew = (A − T)/(A + T); GC-skew = (G − C)/(G + C) [26]. An initial prediction of tRNA genes was performed using the MITOS web server, followed by re-identification utilizing invertebrate mitochondrial codons and default search patterns utilizing the tRNAscan-SE search server (http://lowelab.ucsc.edu/tRNAscan-SE/ (accessed 6 September 2022)) and ARWEN (http://130.235.244.92/ARWEN/ (accessed 6 September 2022)) [27,28].

### 2.4. Phylogenetic Analysis and Gene Order 

The substitution saturation of 12 PCGs in the mitochondrial genome of 70 species was measured using the DAMBE software [29,30]. Based on the results, we used the nucleotide sequences of the 12 PCGs aligned according to default settings using the ClustalW algorithm in MEGA X [25] to construct maximum likelihood (ML) and Bayesian inference (BI) phylogenetic trees. Due to the high divergence, the *atp8* gene was excluded from phylogenetic analysis. The sequences of 12 PCGs of the complete mitogenomes of 70 species from *Mytilidae* (Appendix A) were used for the phylogenetic analysis.

Two Adapedonta species, *Panopea globosa* (NC_025636) and *Panopea abrupta* (NC_033538), were also included in the analysis as an outgroup. The phylogenetic relationships were analyzed using the ML method and constructed in IQ-TREE using the best-fit “GTR + F + R7” model with 1000 nonparametric bootstrapping replicates. The best ML model was selected based on ModelFinder software results [31,32]. The best-fit model (GTR + I + G) for each section was selected by the Akaike Information Criterion (AIC) in MrModeltest 2.3 [33], and then BI analysis was performed using MrBayes 3.2 associating PAUP 4.0 [34] and Modeltest 3.7 software in MrMTgui [34]. The BI analyses were conducted with Markov Chain Monte Carlo (MCMC) using default settings over three independent sets for 2,000,000 generations and were sampled every 1000 steps. The average standard deviation of split frequencies was <0.01, and the first 25% of samples were discarded as burn-in. The resulting phylogenetic tree was visualized through FigTree v1.4.3 (http://tree.bio.ed.ac.uk/software/figtree/ (accessed 6 September 2022)).

### 2.5. Estimation of Divergence Times 

To better investigate the time of fossil differentiation of *Mytilisepta*, we selected 14 *Mytilidae* species, consisting of those belonging to *Brachidontinae*, *Mytilinae*, and *Mytiliseptinae*, to approximate the divergence times. The analysis was conducted with an uncorrelated relaxed clock, the lognormal relaxed molecular clock model, the random starting trees, and the Yule speciation model in BEAST v1.8.4 [35]. We selected two *Mytilidae* fossil calibration points to effectively increase the accuracy of the divergence time estimation. The node calibration points of *Mytilus* were constrained using a normal distribution prior to approximately 78.2 million years ago (Mya) and the roots *Brachidontinae*, *Mytilisepta,* and *Mytilus* were constrained between 334 Mya (http://www.timetree.org/ (accessed 5 March 2023)). Samples were taken from the posterior every 5000 steps for a total of 10,000,000 steps per MCMC run, and then 10% of the steps were discarded by TreeAnnotator v1.8.4 software (https://beast.community/treeannotator#user-interface (accessed 6 September 2022)) after confirming the convergence of the chains using Tracer v.1.6 [36]. The effective sample size of the majority of the parameters was above 200. The divergence times tree was visualized in FigTree v1.4.3 (http://tree.bio.ed.ac.uk/software/figtree/ (accessed 6 September 2022)).

## 3. Results

### 3.1. The Organization and Base Composition of the Genome

The raw sequencing data of the genome were 4854.2 Mb (SRA accession number: SRX19510287), and the clean data were 4831.3 Mb with a GC content of 33.22%; 6374 reads were obtained. The *M. virgata* mitogenome (GenBank accession number: ON193524) was 14,713 bp long. (Figure 1 and Table 1). It contained a common set of 37 mitochondrial genes including 2 rRNA genes, 13 PCGs, and 22 tRNA genes. The distribution of PCGs and RNAs indicated the common pattern of most *Mytilidae* mitogenomes: all 37 mitochondrial genes were encoded on the heavy chain [37,38], as reported for the majority of the Mytilidae spp. The nucleotide compositions were A = 25.75%, T = 43.56%, G = 20.97% and C = 9.73%, A + T = 69.36%, and G + C = 30.7% (Table 1). With an overall nucleotide composition biased toward AT, the *M. virgata* mitogenome suggested that significant strand asymmetry or chain-specific biases can be found in the Mollusca [39]. In addition, the highest A + T content was observed in the *atp8* (73.64%). The A + T content of total PCGs was higher than that of total rRNA and total tRNA genes. 

### 3.2. Use of Protein-Coding Genes and Codons

The total length of 13 PCGs was 11,189 bp (Table 1). The conventional ATG was used as the starting codon in most PCGs, while the initial codons of *cox1*, *cob*, and *atp8* were ATA, ATT, and GTG, respectively. For the termination codons, the typical TAA and TAG were used in most PCGs, but an incomplete termination codon T was found in *cox1* and *atp6* (Table 2). Across the mitogenome, the standard invertebrate mitochondrial genetic code was used for all genetic codons. The codon usage pattern analysis of PCGs showed that the three most frequently detected amino acids in *M. virgata* were Leu (16.97%), Phe (10.96%), and Val (9.70%), and the least common amino acid was Gln (1.17%) (Figure 2). Relative synonymous codon usages for *M. virgata* are summarized in Figure 3 and Table 3; CCU (Pro), UUA (Leu), ACU (Thr), and GCU (Ala) were the four most frequently detected codons, while GGC (Gly) was the least common codon.

### 3.3. Transfer and Ribosomal RNA Genes

Compared to the mitogenomes of most species in the family Mytilidae, the *M. virgata* contained 22 tRNA genes (Figure 4 and Table 1 and Table 2). There were 22 tRNA genes in the mitochondrial genome with a total length of 1435 bp, and the overall A + T content of tRNA genes was 68.50%. In the complete mitogenome of *M. virgata*, the tRNA length varied between 58 and 70 bp. The tRNA genes of *M. virgata* had a negative AT skew (–0.105) and a positive GC skew (0.274). The secondary cloverleaf structure of the 22 tRNAs was investigated and the majority of them were found to have a typical cloverleaf structure, except for *trnS2*, *trnE*, and *trnW*. Strikingly, the TψC loop of the *trnW* gene was completely absent. This characteristic might be a specific feature of tRNA genes in the *M. virgata* mitogenome.

Within this study, the16S rRNA (*rrnL*) and 12S rRNA (*rrnS*) genes of *M. virgata* were identified. The total lengths of the *rrnL* and *rrnS* were 1078 bp and 794 bp, respectively. The rRNAs had an A + T content of 69.61%. Notably, the AT-skew (–0.115) was negative, while the GC-skew (0.321) value of the rRNAs was positive. This indicates that the G content was more prevalent in mitochondrial RNA genes of *M. virgata*.

### 3.4. Gene Arrangement

The gene order of mitogenomes in most *Mollusca* species displayed an extraordinary amount of variation, especially for bivalves [40,41]. Several species of the subfamily *Mytilidae* were chosen as representatives of bivalves to investigate mitochondrial gene rearrangement. The complete mitogenomes sequenced for all species consisted of 12–13 PCGs (some species lacked *atp8* gene), and we manually annotated the sequences that were missing the *atp8* gene (Table 4 and Figure 5); these varied among subfamilies but were generally conserved in closely related species. The majority of species (*Arcuatulinae*, *Mytilinae*, and *Mytiliseptinae*) were identified as lacking the mitochondrial *atp8* gene based on simplification comparisons of the mitochondrial genome structure of the Mytilidae mitogenomes. The gene rearrangement analysis based on genera was preferable because substantial rearrangements still existed within subfamilies of bivalve as we deleted all tRNAs. According to our analysis, the mitochondrial gene order in *Bathymodiolinae* displayed the same gene order arrangement. The subfamily *Modiolinae* contained six *Modiolus* spp. That shared an identical arrangement of 13 PCGs in the order of *cox1*-*nad3*-*atp8*-*atp6*-*nad4*-*cox3*-*nad6*-*nad2*-*cob*-*nad4l*-*nad5*-*cox2*-*nad1*, which was also identical to that of *Bathymodiolinae*. Among the *Xenostrobus secures* and species from *Modiolinae*, both contained *cox1*-*nad3*, *nad4*-*cox3*, and *cox2*-*nad1* gene fragments. Rearrangement events in the mitogenome of *Brachidontinae* species were mainly concentrated in two regions. Compared to *Brachidontes exustus* and *Brachidontes 10haraonic*, *cob* was transposed to between *nad6* and *cox2*, whereas the small fragment *atp6*-*nad1* followed the *atp8* in *Perumytilus purpuratus* and *Geukensia demissa*. The gene order in subfamilies (e.g., *Xenostrobinae* and *Brachidontinae*) retained the small fragments of *nad4*-*cox3*. The gene arrangement in the mitogenome of *M. virgata* was found to be identical to that of *M. keenae* (from *cox1* to *cox2*), and this gene arrangement contained fragments of two other species (*P. purpuratus* and *G. demissa*): the *nad4*-*cox3*-*nad5*-*nad6*-*cob*-*cox2* gene fragment. Additionally, *M. virgata* was found to contain the *atp8* gene, which was consistent with previous studies [3]. For the gene arrangement, the species in *Mytilus* and *Crenomytilus* shared an identical arrangement of *Mytilinae* based on 13 PCGs (in the order of *cox1*-*atp6*-*nad4l*-*nad5*-*nad6*-*cob*-*cox2*-*nad1*-*nad4*-*cox3*-*nad2*-*nad3*-*atp8*), compared with the *Semimytilus patagonicus* and *Mytilus* spp., where there existed the same fragment of *cob*-*cox2*. Gene order differences were observed in the genus Perna, with the fragments spanning from *atp6* to *nad3* and *nad6* to *cox3*, being conserved among *Perna perna*, *Perna canaliculus*, and *Perna viridis*. Surprisingly, the gene sequences of the three genera, *G. demissa*, *P. purpuratus,* and *S. patagonicus,* were found to be identical. Furthermore, the genome order in *Gregariella coralliophaga*, *Crenomytilus grayanus*, and *Mytilus* spp. was almost identical. The most complicated and comprehensive rearrangement of PCGs occurred in *Arcuatulinae*, with a series of relocations of PCGs taking place, affecting the location of most PCGs. Furthermore, within the bivalvia, there were changes in mitogenomes exhibiting genome organization with the following characteristics: transpositions, inversions, and inverse transpositions [42]. As shown in Figure 6, the gene *trnL* moved from between *cox1* and *rrnL* to a position between *nad3* and *rrnS* in *M. virgata*. The transposition of *trnC* from gene block *trnE-trnC-nad3* downstream of *trnQ* occurred in *M. virgata*, as compared with *M. keenae*. The transposition of *trnK* and *trnL*, *trnQ*, *trnH*, and *trnI* occurred in the gene block from *nad1* to *nad4* and made the new gene boundaries *trnK-trnL-trnG, trnQ-trnH-trnN-trnM*, and *trnI-trnT* in *M. virgata*. It was common for mitogenomes to contain two or more copies of a tRNA gene. [43]. For instance, two different *trnM* genes were present in almost all bivalves (e.g., *Mytilus edulis*) [44], but such genes were not identified in *M. virgata*. Analyzing *M. virgata* also revealed a mass of gene arrangements compared to the putative ancestor. Further, the gene orders of the different subfamilies (*Brachidontinae*, *Mytilinae*, and *Mytiliseptinae*) were most distinct from each other.

### 3.5. Phylogenetic Relationships of Mytilidae

In contrast to individual genes or gene fragments, analyzing complete mitochondrial genomes provides a comprehensive view of several genome-level features, including genetic resources, molecular evolution, genome evolution, and phylogeny [45,46]. In the present study, Maximum Likelihood (ML) and Bayesian inference (BI) trees were produced to reconstruct phylogenetic relationships within Pteriomorphia using 12 PCGs based on 70 species (Figure 7). It was anticipated that all phylogenetic analyses with the same topology and approach, but different data matrices, would produce congruent results. Additionally, high support in the majority of nodes between the BI and ML trees was expected for both the data matrices and the tree topologies. The infraclass Pteriomorphia has two clades: the first clade with the family *Mytilidae* and the second clade with families *Pinnidae*, *Margaritidae*, *Pteriidae*, *Pectinidae*, *Ostreidae*, *Arcidae*, and *Cucullaeidae*. The clades of 13 subfamilies were supported with 56–100% ML BP (bootstrap probability) and 0.86–1.0 PP (posterior probability). Within clade 1, *Bathymodiolinae* was well supported (100 BP), but the relationships within it were not well supported. All species in Modiolinae formed a well-supported group. The *X. securis* of the subfamily *Xenostrobinae* was located underneath *Bathymodiolinae*. *Mytilinae* was split into two groups. In the first one, *S. patagonicus* consisted of the subfamily *Mytiliseptinae* and *P. purpuratus*. The genus *Mytilus* spp. (e.g., *M. californanus* and *M. unguiculatus*) and *C. grayanus* were considered to be closely related, forming a group in which all species formed a well-supported clade. The second group, the genus *Perna*, was closely linked with *Arcuatula senhousia* and *Mytella strigata*. Within clade 2, the sister-group relationship of *Pinnidae*, *Margaritidae*, and *Pteriidae* showed high support. *Pectinidae* was the most closely related taxon to these three families. Among the *Ostreidae*, *Ostreinae* was most closely related to *Saccostreinae* and clustered with the *Crassostreinae* species group.

### 3.6. Divergence Times

Retrieving the topology of the maximum clade plausibility tree from the BEAST analysis (Figure 3) yielded the same result as the BI and ML trees in MrBayes (Figure 8). Our estimates show that the tree split into three lineages—*Mytilinae*, *Brachidontinae*, and *Mytiliseptinae*—approximately 333.67 Mya during the early Carboniferous period, which is consistent with the split timing estimated in a previous study [3]. *Mytilus* spp. first diverged approximately 78.79 Mya, with a 95% highest posterior density (HPD) interval spanning the Cretaceous period (61.3–122.0 Mya). The divergence time between *M. virgata*, *M. keenae,* and *P. purpuratus* was estimated to be approximately 116.36 Mya in the late Cretaceous period. Fossil evidence of *M. virgata* and *M. keenae* dates back to 62.65 Mya in the early Paleogene era.

## 4. Discussion

### 4.1. The Organization of the Mitogenome

The complete mitochondrial genome DNA sequence of *M. virgata* contained 37 genes, which is consistent with the findings of the *Mytilidae* family [3]. Among *Mytilisepta*, it was shorter than *M. keenae* (15,902 bp) (Table 1). The variation in mitochondrial genome size is largely attributed to factors such as the frequency of duplicated repeats, horizontal gene transfer, genetic drift, and plasmid-derived regions [47,48,49], but the reason for diversity in the mitochondrial genome length in *Mytilisepta* is unclear. In the *M. virgata* mitogenome, there was a non-coding region between *rrnL* and *trnY* (Figure 1). We consider this non-coding region as the CR/D-loop. The mitogenome was completed immediately following assembly; hence, no further methods were required.

There was a very high prevalence of AT content in mollusca mitogenomes [2,37], with the lowest AT contents typically being over 50% rather than having a balanced nucleotide frequency. GC content in mitochondrial genomes varies among different species, and this variation is attributed to GC-biased gene conversion across the genome. GC content is influenced by mutational bias, the environment, selection, and recombination-associated DNA repair [50,51]. Nucleotide skews are also regularly used to characterize the base compositions of mitogenomes [52] and detect the origin of replication in circular mitogenomes [53]. For *M. virgata*, the lowest AT content was 60.06% (*nad4l*) and the AT skew was negative, while the GC skew was positive, indicating a bias towards G over C. The GC skew was more pronounced, as observed in most studied mollusks, especially bivalves [54,55].

### 4.2. Protein Coding Genes and Codon Usage

According to Ojala et al., incomplete termination codons may result from posttranscriptional modification, in which an “A” is added to the 3′-end of mRNA through polyadenylation to form a complete TAA stop codon, thereby terminating transcription [56]. The explanations for codon usage bias include isoaccepting transfer RNAs, variation in gene expression levels, and GC-content between species [50]. This phenomenon has been observed in other metazoan mitogenomes [57]. Furthermore, in our results, the most frequently used amino acid was Leu, which is consistent with most of the currently studied mitogenomes [58,59].

### 4.3. tRNAs

Following almost all metazoans, the DHU-arm of *trnS2* was reduced to a large DHU loop [60]. The *trnW* completely lacked the TψC loop, as was observed in the *Stenopodidea*, and the complete absence of the TψC loop in *trnR* or *trnM* was occasionally observed in other Mollusca groups [61,62]. A previous study showed that tRNAs commonly exhibited altered structures in mitochondria [63].

### 4.4. Gene Arrangement

Within some phyla of animals, mitochondrial gene arrangement is thought to have undergone only infrequent changes. In the majority of vertebrate mitogenomes, from fish to mammals, the repertoire of coding genes was highly conserved, indicating that no significant gene rearrangements have occurred across different vertebrate clades over a span of 500 million years [64]. On the contrary, gene rearrangements seemed to be more prevalent in invertebrates, where they were accelerated within groups at many taxonomic levels [65,66]. Based on the types of genes rearranged, genome rearrangements can be characterized as minor (tRNAs only) or major (protein-coding and rRNA genes) rearrangements [67]. Various hypotheses have been proposed to explain gene rearrangements in animal mitogenomes: the tandem duplication/random loss (TDRL) model [68], the tandem duplication/non-random loss (TDNL) model [69], the recombination model [70], and the tRNA mispriming model [71]. The TDRL model, which explained the translocation of genes encoded on the same strand through tandem duplication following the random deletion of certain replicated genes, had been broadly applied [71]. Gene arrangements are recognized as powerful evidence for the evolution of organisms and their genomes, and they provide explanations that resolve the relationships between distant lineages in terms of phylogenetic signals [40].

The phenomenon of missing *atp8* is common in bivalves (e.g., species in *Mactridae*, *Arcidae*, and *Pectinidae*) [1,55,62,72,73]. Some researchers have proposed that this peptide possibly has a dispensable function in the ATPase complex, either because the gene is transferred to the nucleus or because the *atp8* protein is too short and variable in length to be annotated [1,38]. A significant number of transpositions and inversions have been identified in comparison to the putative ancestral sequence. Thus far, one assumption that explains why the bivalve gene order is so variable has been proposed: there may be occasional failures in the unique mtDNA inheritance pattern (DUI) machinery, which may also involve gene translocations, gene duplication/lost events, and recombination [40]. Lubośny et al. proposed that bivalvia was considered to be lacking *atp8* because of the poor similarity between protein-coding gene sequences in genetically closed species. It was, in fact, present but highly divergent and just not annotated in some of them [74]. Additionally, we manually annotated *atp8* in all those species that lack sequences with *atp8*. The outcome of the manual annotation process was found to be highly consistent with the findings reported in the previous article.

### 4.5. Phylogeny

Recent molecular phylogenetic studies have proposed different hypotheses for the higher-level phylogeny of Pteriomorphia. However, most of the families within this order remain unresolved [3,75]. Given this, the taxonomy of the *Mytilidae* family is primarily reliant on morphological characteristics, and there is little consensus regarding specific taxonomic assignments within the family. Additional studies are, therefore, needed to address phylogenetic relationships within the *Mytilidae* family.

In our study, the subfamilies *Bathymodiolinae* and *Modiolinae* showed sister relationships, which were identical both to the topological structure of the phylogenetic tree based on nuclear *18S rRNA* and mitochondrial *COI* genes constructed for all species in the deep sea and to the results based on transcriptome sequences of representative members of the *Mytilidae* [76]. This is also consistent with the results based on transcriptome sequences of representative members of *Mytilidae* [77], but different from the result based on *18S rRNA* variability, where *Modiolinae* occupied a position under the *Mytilinae* [78]. The apparent monophyly of *Modiolinae* differs from the previous results presented by Distel [78]; additionally, in our dataset, *X. secures* (*Limnoperna fortunei*) of the subfamily *Xenostrobinae* was located underneath the *Bathymodiolinae*, which is consistent with the previous study [76]. The findings for the *Perna* species were the same as that of Wood et al. [79] and Cunha et al. [80], who found evidence for the monophyly of the *Perna* genus and built the phylogenetic trees without using the sequences from the *Brachidontes* genus. Combosch et al. validated the monophyly of the genus *Perna* and indicated a misidentification in the previous study [81]. In our analysis, the *Mytilisepta* was clustered with *P. purpuratus*, as in previous studies [18,82], but this finding differed from the results presented by Zhao et al. [83]. Moreover, Combosch et al. concluded that *M. virgata* was most closely related to *B. exustus* by using a five-gene Sanger-based approach [81]. In addition, early phylogeny based on the mitochondrial gene *cox1* suggested that *Mytilisepta* was a monophyletic genus placed within *Mytilidae* [84,85], as confirmed by our results. It was confirmed that the genus *Mytilus* spp. (e.g., *M. californanus* and *M. unguiculatus*) and *C. grayanus* were closely related and formed one clade. *C. grayanus* was classified as a *Mytilus* by Dunker (World Register of Marine Species, WORMS), which indicated that *C. grayanus* could be closely related to *M. californanus* and *M. unguiculatus*. The phylogenetic tree showed five subfamilies of *Brachidontinae*, *Mytilinae*, *Mytiliseptinae*, *Musculinae*, and *Arcuatulinae* (*A. senhousia* and *M. strigata*) classified into one clade. There has long been some debate regarding the branch of the *Perna* and *Arcuatula* (*Musculista*). Earlier studies based on spermatozoa structure and shell morphology placed them into *Mytilinae* and *Musculinae*, respectively [86,87]. Subsequently, both *Perna* and *Arcuatula* were located in the same subfamily (*Musculinae*) because of the anatomical feature of the pericardial complex being located between two posterior byssal retractor muscle blocks [3]. In our study, the relationships between *A. senhousia* and *Perna* (*P. canaliculus* and *P. viridis*) strengthen the previously established connections between *Arcuatula* and *Perna* [3].

In the second clade, the family Pectinidae was nested within the *Pinnidae*, *Margaritidae*, and *Pteriidae*, forming a closer relationship. This result differs from those of previous studies that used *cox1* and *18S rDNA* and suggested that *Pectinidae* was closer to Arcidae than any other family [88,89]. The phylogenetic trees, however, support a closer relationship between *Ostreidae* and *Arcidae*, with high support values, contradicting the analysis by Sun and Gao [90]. In our study, we observed that *Ostreinae* and *Saccostreinae* were clustered together, with the result that the three genera (*Ostreinae*, *Saccostreinae*, and *Crassostreinae*) formed a monophyletic group with strong support [91].

### 4.6. Divergence Times

In contrast to our findings, the divergence time estimated for the *Austromytilus* + *Mytilisepta* + *Perumytilus* clade was relatively insensitive to the prior selection and was estimated to be 13.35 Mya under the Yule prior [86]. The Paleobiology Database (https://paleobiodb.org (accessed 6 September 2022)) indicates that the oldest *Mytilisepta* fossil was from the late or upper Eocene. It was beyond the scope of this paper to suggest that further analysis might be needed to produce an affordable taxonomic revision. To our surprise, there were discrepancies between our findings and those of Zhao et al. Therefore, further in-depth investigations should be conducted in the future [83].

## 5. Conclusions

In our study, we used the next-generation sequencing method to sequence the mitogenome of *M. virgata*, which was 14,713 bp in length. Our analyses show that all 37 mitochondrial genes are encoded on the heavy chain, with a negative AT skew and a positive GC skew in the total mitochondrial genome. Among the tRNA secondary structures, only *trnS2* lacked DHU stems, while the TψC loop of the *trnW* gene was entirely absent, and the *trnE* was missing the anticodon arm. Furthermore, gene rearrangements were especially apparent among the *Mytiloidea* mitogenomes. Our analysis of the mitochondrial genome provides further support for the previous elevation of the family *Mytilidae* to the order level. Although there is currently a lack of molecular data on *M. virgata*, comprehensive data on the taxa of *Mytiliseptinae* was available to enhance our understanding of the rearrangements, evolutionary events, and phylogenetic position of the *Mytilidae*.

## Figures and Tables

**Figure 1 genes-14-00910-f001:**
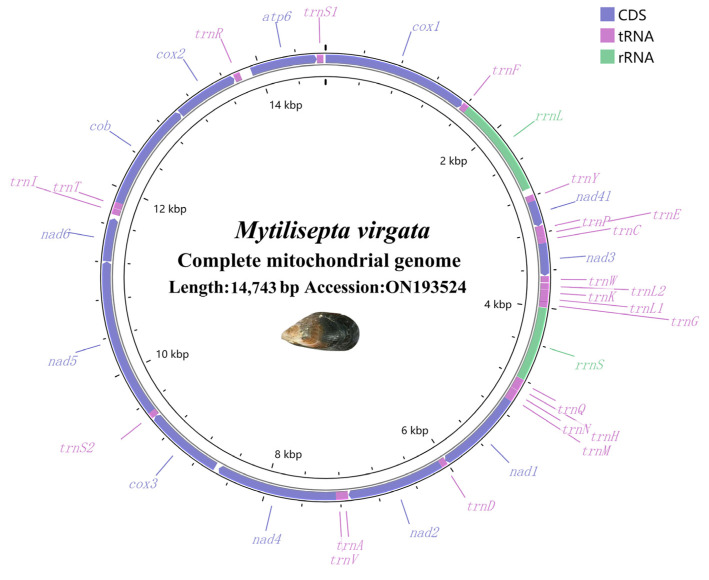
Circular mitogenome map of the mitochondrial genome of *M. virgata*. Protein coding, ribosomal, and tRNA genes are shown with standard abbreviations. Arrows indicate the orientation of gene transcription.

**Figure 2 genes-14-00910-f002:**
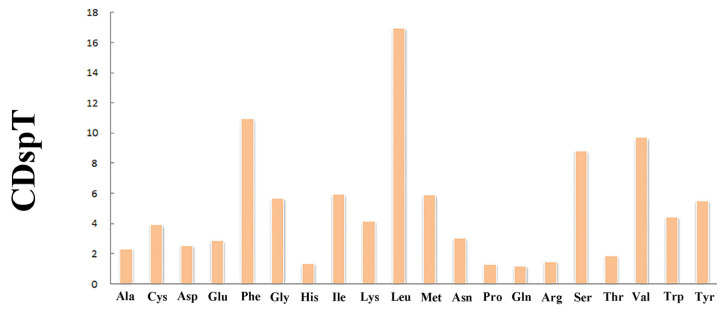
Amino acid compositions of *M. virgata* mitochondrial genomes.

**Figure 3 genes-14-00910-f003:**
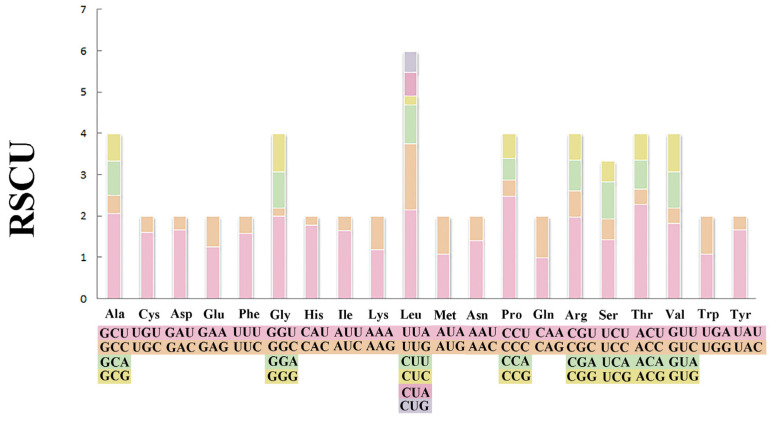
Relative synonymous codon usages (RSCU) in the mitogenomes of *M. virgata*.

**Figure 4 genes-14-00910-f004:**
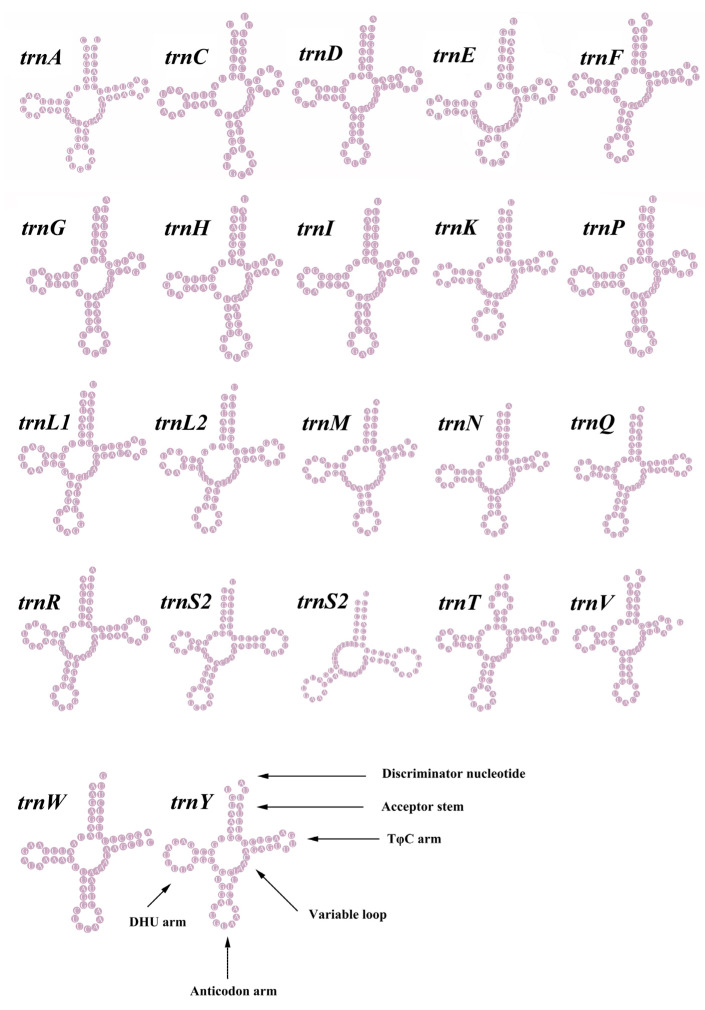
Putative secondary structures of the tRNA genes in the mitogenome of *M. virgata*. The tRNAs are labeled with the abbreviations of their corresponding amino acids.

**Figure 5 genes-14-00910-f005:**
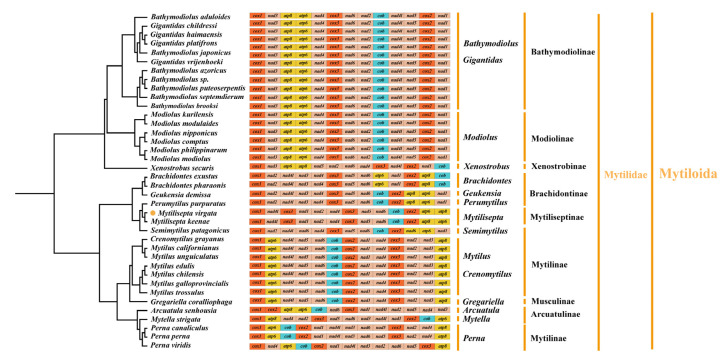
Linearized representation of the mitochondrial gene arrangement in *Mytilidae* bivalves (see Figure 7 for detailed relationships among *Mytilidae* species). Gene segments are not drawn to scale. The gene arrangement of all genes is transcribed from left to right. The orange dot indicates the specie of this study.

**Figure 6 genes-14-00910-f006:**
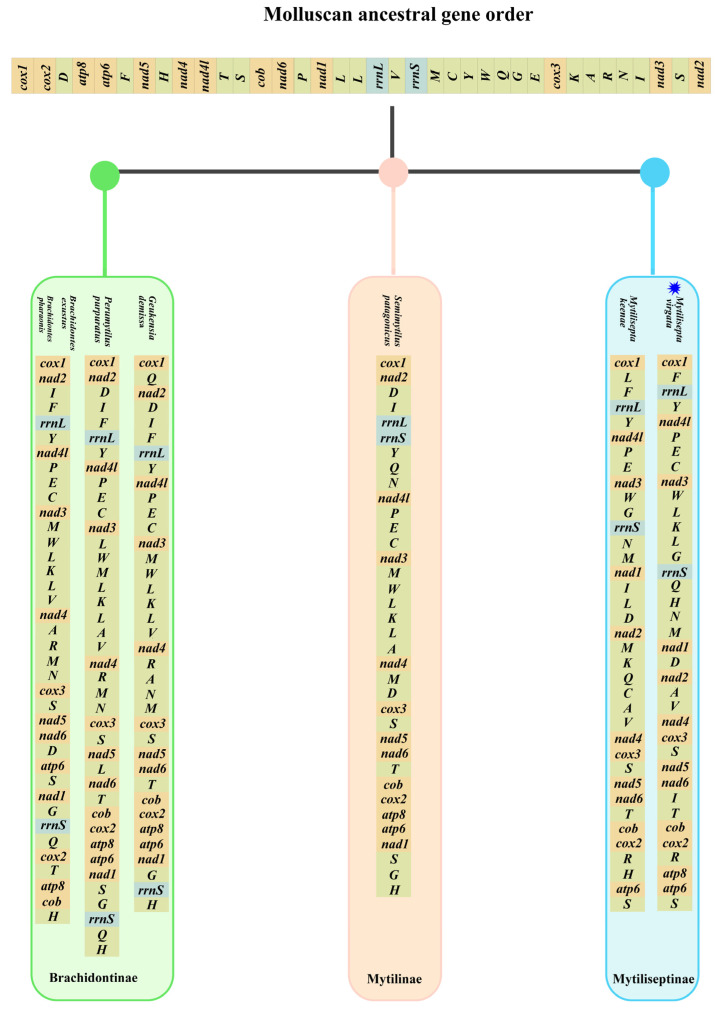
Comparison of mitochondrial gene rearrangements between the *M. virgata* and Molluscan ancestral gene orders. Gene segments are not drawn to scale. The blue symbol indicates the specie of this study.

**Figure 7 genes-14-00910-f007:**
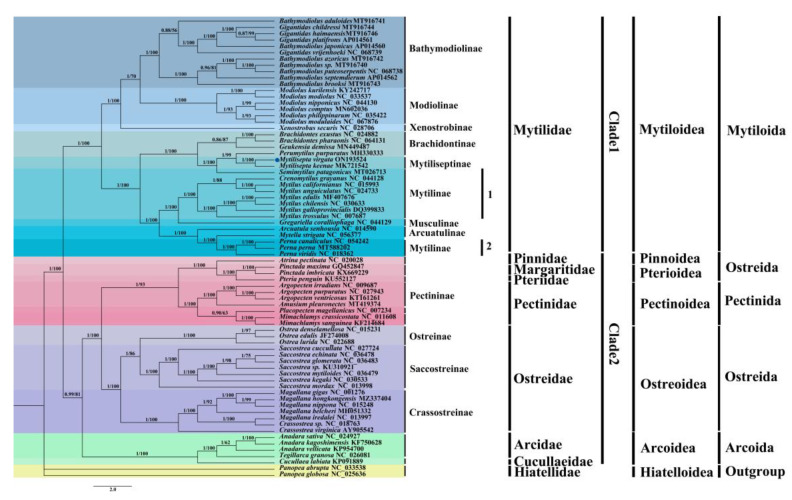
The phylogenetic tree for *M. virgata* and other bivalvia species based on 12 PCGs. Phylogenetic tree inferred using Bayesian inference (BI) and maximum likelihood (ML) methods. The value on the left side of the slash is the posterior probabilities estimated by the Bayesian tree, and the value on the right side is the maximum likelihood tree. The blue dot indicates *M. virgata* in this study.

**Figure 8 genes-14-00910-f008:**
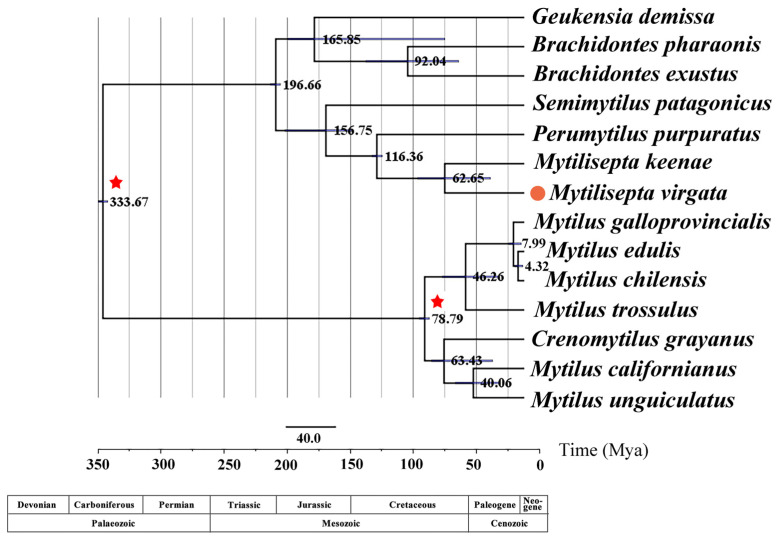
Divergence time estimations for *Brachidontinae*, *Mytilinae*, and *Mytiliseptinae* by using Bayesian relaxed dating methods (BEAST) based on the nucleotide sequences of 12 PCGs (except *atp8*). Horizontal bars indicate 95% relevant nodes and credible intervals of the estimated divergence time. Calibration taxa are indicated with red asterisks on the corresponding nodes, and the orange dot indicates *M. virgate* in this study.

**Table 1 genes-14-00910-t001:** Skewness of the *M. virgata* mitogenome.

Region	Size (bp)	A (%)	T (%)	G (%)	C (%)	A + T (%)	AT-Skew	GC-Skew
Mitogenome	14,713	25.75	43.56	20.97	9.73	69.31	–0.257	0.366
*cox1*	1564	22.89	43.48	22.06	11.57	66.37	–0.310	0.312
*cox2*	687	24.89	41.48	22.56	11.06	60.70	–0.250	0.342
*atp6*	715	20.70	47.69	21.96	9.65	63.96	–0.395	0.389
*atp8*	122	20.93	52.71	23.26	3.10	73.64	–0.431	0.765
*cox3*	843	23.61	45.55	21.83	9.02	69.16	–0.317	0.415
*nad3*	348	23.28	47.70	20.69	8.33	70.98	–0.344	0.426
*nad1*	945	24.02	45.29	21.27	9.42	69.31	–0.307	0.386
*nad5*	1710	25.32	44.27	22.81	7.60	64.96	–0.272	0.500
*nad4*	1302	24.04	47.85	20.20	7.91	71.89	–0.331	0.437
*nad4l*	273	26.74	42.86	22.71	7.69	69.60	–0.232	0.494
*nad6*	465	21.51	50.75	21.08	6.67	62.42	–0.405	0.519
*cob*	1167	23.31	45.59	19.37	11.74	64.05	–0.323	0.245
*nad2*	1041	26.61	43.04	19.88	10.47	63.76	–0.236	0.310
tRNAs	1435	30.66	37.84	20.07	11.43	68.50	–0.105	0.274
rRNAs	1872	30.78	38.83	20.08	10.32	69.61	–0.115	0.321
PCGs	11,189	23.94	45.27	21.36	9.43	69.21	–0.597	0.387

**Table 2 genes-14-00910-t002:** Annotation of the *M. virgata* mitochondrial genome.

Gene	Strand	Location		Length	Codons	Intergenic Nucleotide (bp)	Anticodon
		Start	Stop				
*cox1*	+	1	1564	1564	ATA/T	0	
*trnF*	+	1565	1631	67		0	TTC
*rrnL*	+	1632	2709	1078		73	
*trnY*	+	2783	2850	68		0	TAC
*nad4l*	+	2851	3123	273	ATG/TAA	0	
*trnP*	+	3124	3189	66		0	CCA
*trnE*	+	3190	3247	58		1	GAA
*trnC*	+	3249	3312	64		0	TGC
*nad3*	+	3313	3660	348	ATG/TAA	5	
*trnW*	+	3666	3730	65		10	TGA
*trnL2*	+	3741	3805	65		2	TTA
*trnK*	+	3808	3873	66		0	AAA
*trnL1*	+	3874	3939	66		−4	CTA
*trnG*	+	3936	3997	62		5	GGA
*rrnS*	+	4003	4796	794		3	
*trnQ*	+	4800	4867	68		0	CAA
*trnH*	+	4868	4928	61		3	CAC
*trnN*	+	4932	4996	65		0	AAC
*trnM*	+	4997	5059	63		1	ATG
*nad1*	+	5061	6005	945	ATG/TAG	0	
*trnD*	+	6004	6071	68		1	GAC
*nad2*	+	6073	7113	1041	ATG/TAA	5	
*trnA*	+	7119	7184	66		0	GCA
*trnV*	+	7185	7247	63		0	GTA
*nad4*	+	7248	8549	1302	ATG/TAA	42	
*cox3*	+	8592	9434	843	ATT/TAA	0	
*trnS2*	+	9435	9497	63		0	TCA
*nad5*	+	9498	11,207	1710	ATG/TAG	3	
*nad6*	+	11,211	11,675	465	ATG/TAA	39	
*trnI*	+	11,715	11,779	65		3	ATC
*trnT*	+	11,783	11,849	67		0	ACA
*cob*	+	11,850	13,016	1167	ATT/TAG	5	
*cox2*	+	13,022	13,708	687	ATG/TAG	5	
*trnR*	+	13,714	13,783	70		1	CGA
*atp8*	+	13,785	13,913	122	GTG/TAA	−7	
*atp6*	+	13,907	14,621	715	ATG/T	0	
*trnS1*	+	14,622	14,690	69		23	AGC

**Table 3 genes-14-00910-t003:** The codon number and relative synonymous codon usage in the mitochondrial genomes of *M. virgata*. The asterisk (*) in the table indicates the stop codon.

Codon	Count	RSCU	Codon	Count	RSCU	Codon	Count	RSCU	Codon	Count	RSCU
UUU (F)	399	1.58	UCU (S)	73	1.43	UAU (Y)	213	1.67	UGU (C)	145	1.6
UUC (F)	106	0.42	UCC (S)	26	0.51	UAC (Y)	42	0.33	UGC (C)	36	0.40
UUA (L)	282	2.16	UCA (S)	46	0.90	UAA (*)	172	1.16	UGA (W)	112	1.09
UUG (L)	206	1.58	UCG (S)	25	0.49	UAG (*)	124	0.84	UGG (W)	93	0.91
CUU (L)	124	0.95	CCU (P)	38	2.49	CAU (H)	56	1.78	CGU (R)	34	1.97
CUC (L)	29	0.22	CCC (P)	6	0.39	CAC (H)	7	0.22	CGC (R)	11	0.64
CUA (L)	74	0.57	CCA (P)	8	0.52	CAA (Q)	27	1.00	CGA (R)	13	0.75
CUG (L)	67	0.51	CCG (P)	9	0.59	CAG (Q)	27	1.00	CGG (R)	11	0.64
AUU (I)	226	1.65	ACU (T)	49	2.28	AAU (N)	98	1.41	AGU (S)	88	1.73
AUC (I)	48	0.35	ACC (T)	8	0.37	AAC (N)	41	0.59	AGC (S)	21	0.41
AUA (M)	147	1.08	ACA (T)	15	0.70	AAA (K)	114	1.19	AGA (S)	69	1.36
AUG (M)	125	0.92	ACG (T)	14	0.65	AAG (K)	77	0.81	AGG (S)	59	1.16
GUU (V)	203	1.82	GCU (A)	55	2.06	GAU (D)	98	1.68	GGU (G)	130	1.99
GUC (V)	41	0.37	GCC (A)	12	0.45	GAC (D)	19	0.32	GGC (G)	13	0.20
GUA (V)	100	0.89	GCA (A)	22	0.82	GAA (E)	83	1.26	GGA (G)	58	0.89
GUG (V)	103	0.92	GCG (A)	18	0.67	GAG (E)	49	0.74	GGG (G)	60	0.92

**Table 4 genes-14-00910-t004:** Annotation of *atp8* gene in Mytilidae.

Species	Position (bp)	Size (bp)	Intergenic Region (bp)	Start Codon	Stop Codon	GenBank
*Modiolus kurilensis*	676–861	186	191	ATG	TAA	KY242717
*Modiolus modiolus*	3240–3419	180	194	ATG	TAA	NC_033537
*Modiolus nipponicus*	3409–3636	228	304	ATG	TAA	NC_044130
*Modiolus philippinarum*	16,113–16,304	192	209	ATG	TAA	NC_035422
*Modiolus comptus*	3106–3276	171	670	ATG	TAG	MN602036
*Arcuatula senhousia*	7403–7594	192	216	ATG	TAG	NC_014590
*Mytilus unguiculatus*	8677–8949	273	286	ATG	TAA	NC_024733
*Mytilisepta keenae*	15,718–15,882	165	186	ATG	TAA	MK721542

## Data Availability

The data that support the findings of this study are available from the corresponding author upon reasonable request.

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
