# Peer review of "The Complete Mitochondrial Genome of *Mytilisepta virgata* (Mollusca: Bivalvia), Novel Gene Rearrangements, and the Phylogenetic Relationships of Mytilidae"

_genes, 2023, doi:10.3390/genes14040910_

Round 1

Reviewer 1 Report

The manuscript is well deserved for publication in this journal. Nevertheless, a few minor revisions are needed. The suggestions revisions are as below.

1- Page number 57 introduction section, "Mytilisepta bifurcata" it needs to be as M.bifurcata

2- Page 60 " spots" should be replaced with any other suitable word.

3-Page 74 "barely" should be replaced with any other suitable word.

4- Page 93 " Either the samples were stored in 95% ethonol or in 99.9% absolute ethanol.

5- I suggest to add a little elaboration of higher and lower amino acid composition in the discussion section.

6- The font size of Figure 7 needs to be improved .

Author Response

Dear Editors and Reviewers:

Thank you for your comments on the paper. Efforts have been made to clarify the following points and correct the mistakes. The modifications based on the suggestions are explained in further detail below.

  • Page number 57 introduction section, "Mytilisepta bifurcata" it needs to be as bifurcata

Answer: Thanks for your advice. We have corrected it.

2- Page 60 " spots" should be replaced with any other suitable word.

Answer: Thanks for your advice. We have changed “spots” to “locations”.

3-Page 74 "barely" should be replaced with any other suitable word.

Answer: Thanks for your advice. We have changed “barely” to “rarely”.

4- Page 93 " Either the samples were stored in 95% ethonol or in 99.9% absolute ethanol.

Answer: We have corrected it to “95% ethanol”.

5- I suggest to add a little elaboration of higher and lower amino acid composition in the discussion section.

Answer: Thanks for your advice. We have added it to “Protein coding genes and codon usage”.

6- The font size of Figure 7 needs to be improved.

Answer: Thanks for your advice. We have improved it.

Reviewer 2 Report

The text was very difficult to read, in particular in the results and discussion sections. It will need to be completely overhauled as the revision of the content was very difficult to for this reason. 

For what I can interpret the work is fairly solid, and only the main text needs intervention.

Moreover, at row 104-105 the use of Illumina NovaSeq X Ten platform is stated, but I think such machine does not exist, could you please check this?

Table 1 is very big, it should be moved to supplementary data

at row 172-173, the first two sentences about the composition seem to be contradicting each other.

At row 182 the number of PCGs is reported as 13, but everywhere in the text it is 12, please check.

Starting from the results section the names of the species are never italicized as they should, the same should be done for the names of the genes.

Author Response

Dear Editors and Reviewers:

Thank you for your comments on the paper. Efforts have been made to clarify the following points and correct the mistakes. The modifications based on the suggestions are explained in further detail below.

1.Moreover, at row 104-105 the use of Illumina NovaSeq X Ten platform is stated, but I think such machine does not exist, could you please check this?

Answer: Thanks for your advice. We have corrected it to “Illumina NovaSeq 6000 platform”.

2.Table 1 is very big, it should be moved to supplementary data

Answer: Thanks for your advice. We have corrected it.

3.at row 172-173, the first two sentences about the composition seem to be contradicting each other.

Answer: Thanks for your advice. We have corrected the first two sentences.

4.At row 182 the number of PCGs is reported as 13, but everywhere in the text it is 12, please check.

Answer: Thanks for your advice. We have corrected it.

5.Starting from the results section the names of the species are never italicized as they should, the same should be done for the names of the genes.

Answer: Thanks for your advice. We have corrected it.

Reviewer 3 Report

In this manuscript authors generated the complete sequence of mitochondrial genome Mytilisepta virgata. Authors characterized the genome sequence and clarified the gene structure, base composition, and codon usage. The gene arrangement was inferred together with other Mytilidae bivalves and phylogenetic analyses were carried out.

Page 3, 1st sentence of the 3rd paragraph. “12 PCGs” -> “12 protein-coding genes (PCGs)”

Page 3, 3rd sentence of the 3rd paragraph.  “the PCGs which aligned” -> “the PCGs aligned”

Page 3, 4th sentence of the 4th paragraph.  “analysed by ML and constructed in IQ-TREE using GTR+F+R7 model and the best-fit substitution model with ModelFinder. The best-fit model (GTR+I+G) for each section was selected” Which was the best-fit model, GTR+F+R7 or GTR+I+G?

Page 4, line 9. “88 Mya” Please add a reference.

Page 14, line 3. “BP and 0.81-1.0 PP” -> “BP(bootstrap probability) and 0.81-1.0 PP (posterior probability)”

Page 14, line 4. “(100B)” -> “(100 BP)”?

Page 14, line 5. “The L. fortune of the subfamily Arcuatulinae underneath the Bathymodiolinae.” The meaning is not clear.

Page 14, line 8. “i.g.” -> “e.g.”?

Figure 1. It is better to use larger fonts and darker color for characters outside the circle.

Author Response

Dear Editors and Reviewers:

Thank you for your comments on the paper. Efforts have been made to clarify the following points and correct the mistakes. The modifications based on the suggestions are explained in further detail below.

1.Page 3, 1st sentence of the 3rd paragraph. “12 PCGs” -> “12 protein-coding genes (PCGs)”

Answer: Thanks for your advice. We have corrected it to “12 protein-coding genes (PCGs)”.

2.Page 3, 3rd sentence of the 3rd paragraph. “the PCGs which aligned” -> “the PCGs aligned”

Answer: Thanks for your advice. We have corrected it to “the PCGs aligned”.

3.Page 3, 4th sentence of the 4th paragraph. “analysed by ML and constructed in IQ-TREE using GTR+F+R7 model and the best-fit substitution model with ModelFinder. The best-fit model (GTR+I+G) for each section was selected” Which was the best-fit model, GTR+F+R7 or GTR+I+G?

Answer: Thanks for your advice. The “GTR+F+R7” is the best model is the best model for building ML tree and the “GTR+I+G” is the best model is the best model for building BI tree.

4.Page 4, line 9. “88 Mya” Please add a reference.

Answer: Thanks for your advice. We have added it.

Page 14, line 3. “BP and 0.81-1.0 PP” -> “BP (bootstrap probability) and 0.81-1.0 PP (posterior probability)”

Answer: Thanks for your advice. We have corrected it

5.Page 14, line 4. “(100B)” -> “(100 BP)”?

Answer: Thanks for your advice. We have corrected it.

6.Page 14, line 5. “The L. fortune of the subfamily Arcuatulinae underneath the Bathymodiolinae.” The meaning is not clear.

Answer: Thanks for your advice. We have rewritten this sentence.

7.Page 14, line 8. “i.g.” -> “e.g.”?

Answer: Thanks for your advice. We have corrected it.

  1. Figure 1. It is better to use larger fonts and darker color for characters outside the circle.

Answer: Thanks for your advice. We have corrected it.

Reviewer 4 Report

Review (Genes): The complete mitochondrial genome of Mytilisepta virgata (Mollusca: Bivalvia), novel gene rearrangements and phylogenetic relationships of Mytilidae

1. The mitogenome sequence for Mytilisepta virgata has been already published (Genbank MK721548) and the authors (you) even cite it. There is no novelty aspect in this study.

2. The Atp8 gene for this species has been found. (Genbank MK721548) Lee Y, Kwak H, Shin J, Kim S-C, Kim T, Park J-K. 2019. A mitochondrial genome phylogeny of Mytilidae (Bivalvia: Mytilida). Molecular Phylogenetics and Evolution 139:106533. DOI: 10.1016/j.ympev.2019.106533.

3. The Perna perna controversies have been already resolved.

Cunha RL, Nicastro KR, Zardi GI, Madeira C, McQuaid CD, Cox CJ, Castilho R. 2022. Comparative mitogenomic analyses and gene rearrangements reject the alleged polyphyly of a bivalve genus. PeerJ 10:e13953. DOI: 10.7717/peerj.13953.

Lubośny M, Śmietanka B, Arculeo M, Burzyński A. 2022. No evidence of DUI in the Mediterranean alien species Brachidontes pharaonis (P. Fisher, 1870) despite mitochondrial heteroplasmy. Scientific Reports 12:8569. DOI: 10.1038/s41598-022-12606-6.

4. The length of the 16S rRNA annotation is wrong (to short).

5. The article is very difficult to read. Sentences are not grammatically correct. Language needs to be improved.

6. I understand that publication is a part of the Master thesis in Asia, but the thesis supervisor has to make sure that the quality of the article is good enough to be sent to journals.

7. (Lines 72-24): This is not true (look: MK721548 and KX094521)

8. Missing accession number for NGS sequencing reads.

9. In some places there is no italicisation for species names and genes.

10. Few double spaces in the text.

11. It seems like phylogenetic analyses were performed for concatenated mitochondrial genes. Maybe it would be better to run genes separately with different models. IQ-TREE has this option. The bootstrap value 1000 is quite low.

12. No word on doubly uniparental inheritance (DUI) in Mytilidae? Was your individual male or female? If you had used gonad tissue from a male individual there would be an occasion to check this species for DUI.

13. There is an approximately 8.6-8.8% divergence between M. virgata from China and the one from Korea. For me it is something interesting enough to write about.

14. It would be nice to Cite authors of all GenBank records you use in your phylogenetic reconstruction (only recommendation).

15. (line 183): In your GenBank record cox1 (italicised if you are writing about gene, nonitalicised if about protein) starts with ATA not CGA (CGA is not a start codon in inv_mtDNA genetic code).

16. “The conventional ATA was used as the start codon in most PCGs” and then in table 3 we have 9 time ATG and 1 ATA and 2 ATT. Something is wrong in the text.

17. There is more, but English needs to be corrected first… 

You should re-think the whole concept of this article. Find another angle to approach it.

Author Response

Dear Editors and Reviewers:

Thank you for your comments on the paper. Efforts have been made to clarify the following points and correct the mistakes. The modifications based on the suggestions are explained in further detail below.

  1. The mitogenome sequence for Mytilisepta virgata has been already published (Genbank MK721548) and the authors (you) even cite it. There is no novelty aspect in this study.

Answer: Thanks for your advice. The previous sequence (accession no. MK721548) could not be looped due to some problems and was an unverified mitochondrion sequence, now we did it again using next-generation sequencing technology, the results showed that the mitochondrial sequence formed a loop structure and the sequence has been verified by NCBI.

  1. The Atp8 gene for this species has been found. (Genbank MK721548) Lee Y, Kwak H, Shin J, Kim S-C, Kim T, Park J-K. 2019. A mitochondrial genome phylogeny of Mytilidae (Bivalvia: Mytilida). Molecular Phylogenetics and Evolution 139:106533. DOI: 10.1016/j.ympev.2019.106533.

Answer: Thanks for your advice. The previous sequence (accession no. MK721548) eventually could not be looped and was an unverified mitochondrion sequence. ATP8 deficiency does exist in many species in the Mytilidae (e.g. Modiolus kurilensis, Mytilus californianus, Perna perna) (Lee Y, Kwak H, Shin J, Kim SC, Kim T, Park JK. A mitochondrial genome phylogeny of Mytilidae (Bivalvia: Mytilida). Mol Phylogenet Evol. 2019 Oct; 139:106533. doi: 10.1016/j.ympev.2019.106533.)

  1. The Perna perna controversies have been already resolved.

Answer: Thanks for your advice. We have rewritten this part.

  1. The length of the 16S rRNA annotation is wrong (to short).

Answer: Thanks for your advice. We had re-annotated and corrected the rrnl to ensure it was accurate.

  1. The article is very difficult to read. Sentences are not grammatically correct. Language needs to be improved.

Answer: Thanks for your advice. We have improved our language.

  1. I understand that publication is a part of the Master thesis in Asia, but the thesis supervisor has to make sure that the quality of the article is good enough to be sent to journals.

Answer: Thanks for your advice. We have improved our language.

  1. (Lines 72-24): This is not true (look: MK721548 and KX094521)

Answer: Thanks for your advice. We have deleted this sentence.

  1. Missing accession number for NGS sequencing reads.

Answer: Thanks for your advice. We have submitted the data and the accession number is SRX19510287.

  1. In some places there is no italicisation for species names and genes.

Answer: Thanks for your advice. We have corrected this.

  1. Few double spaces in the text.

Answer: Thanks for your advice. We have corrected this.

  1. It seems like phylogenetic analyses were performed for concatenated mitochondrial genes. Maybe it would be better to run genes separately with different models. IQ-TREE has this option. The bootstrap value 1000 is quite low.

Answer: Thanks for your advice. We build the IQ-TREE based on the reference (Lam-Tung, N., Schmidt, H.A., Arndt, V.H., Quang, M. B. Iq-Tree: a fast and effective stochastic algorithm for estimating maximum-likelihood phylogenies. Mol. Phylogenet. Evol. 2015, 32, 268-274. https://doi.org/10.1093/molbev/msu300)

  1. No word on doubly uniparental inheritance (DUI) in Mytilidae? Was your individual male or female? If you had used gonad tissue from a male individual there would be an occasion to check this species for DUI.

Answer: Thanks for your advice. We have added the content of doubly uniparental inheritance (DUI) and the we extracted total genomic DNA by the rapid salting-out method from the adductor muscle, we will used gonad tissue from a male individual to check this species for DUI next time.

  1. There is an approximately 8.6-8.8% divergence between M. virgata from China and the one from Korea. For me it is something interesting enough to write about.

Answer: Thanks for your advice. We primarily describe the mitochondrial genome of M. virgata about gene rearrangement and phylogeny compared with other species in Mytilidae and we will add this content in relevant article next time.

  1. It would be nice to Cite authors of all GenBank records you use in your phylogenetic reconstruction (only recommendation).

Answer: Thanks for your advice. We will improve next time.

  1. (line 183): In your GenBank record cox1 (italicised if you are writing about gene, nonitalicised if about protein) starts with ATA not CGA (CGA is not a start codon in inv_mtDNA genetic code).

Answer: Thanks for your advice. We have corrected it.

  1. “The conventional ATA was used as the start codon in most PCGs” and then in table 3 we have 9-time ATG and 1 ATA and 2 ATT. Something is wrong in the text.

Answer: Thanks for your advice. We have corrected it.

  1. There is more, but English needs to be corrected first…

Answer: Thanks for your advice. We have improved our language.

Round 2

Reviewer 2 Report

Dear authors,

you really need to check the english over the entire manuscript, it is very hard to understand and reviewers are not proofreaders, nor they have the time to read sentences three times to make sense of them.

In the first round of review I already highlighted this issue, but you ignored my request of improvement. I still have a very hard time in giving a complete opinion about the content of your manuscript, so I do not feel confident on my review yet.

As I suggested you moved the table with the accession ids to supplementary material, but the reference to it is in the wrong place, as it should be in the materials section after you mention the usage of deposited data for the first time. On this matter:

- The entry NC_013998 belongs to Saccostrea mordax, not Saccostrea scyphophylla (https://www.marinespecies.org/aphia.php?p=taxdetails&id=138300)

- Pinctada fucata is the accepted name of Pinctada martensii and you reported the accession id of the mitogenome of Pinctada imbricata

Moreover, I did not notice this the first time, you discussed the peculiar position of Perna perna in your phylogenetic analysis, but a quick blast search reveals that the uploader of the genome misclassified it, luckily there is another genome with the id MT588202.1. You would need to repeat the analysis and change the manuscript accordingly as you are likely using the wrong reference for this species. This is also reflected in the gene rearrangement part

When you state a probability it is either comprised between 0 and 1 or it is a percentage. In the latter case you need to explicit the % sign

Author Response

Dear Editors and Reviewers:

Thank you for your comments on the paper. Efforts have been made to clarify the following points and correct the mistakes. The modifications based on the suggestions are explained in further detail below.

1.you really need to check the english over the entire manuscript, it is very hard to understand and reviewers are not proofreaders, nor they have the time to read sentences three times to make sense of them. In the first round of review, I already highlighted this issue, but you ignored my request of improvement. I still have a very hard time in giving a complete opinion about the content of your manuscript, so I do not feel confident on my review yet.

Answer: Thanks for your advice. We have improved our language.

2.As I suggested you moved the table with the accession ids to supplementary material, but the reference to it is in the wrong place, as it should be in the materials section after you mention the usage of deposited data for the first time.

Answer: Thanks for your advice. We have corrected it.

  1. On this matter: The entry NC_013998 belongs to Saccostrea mordax, not Saccostrea scyphophylla (https://www.marinespecies.org/aphia.php?p=taxdetails&id=138300)

Answer: Thanks for your advice. We have corrected it.

5.Pinctada fucata is the accepted name of Pinctada martensii and you reported the accession id of the mitogenome of Pinctada imbricata

Answer: Thanks for your advice. We have corrected it.

6.Moreover, I did not notice this the first time, you discussed the peculiar position of Perna perna in your phylogenetic analysis, but a quick blast search reveals that the uploader of the genome misclassified it, luckily there is another genome with the id MT588202.1. You would need to repeat the analysis and change the manuscript accordingly as you are likely using the wrong reference for this species. This is also reflected in the gene rearrangement part

Answer: Thanks for your advice. We have rebuilt the tree used the sequence (MT588202.1) and rewritten this part.

7.When you state a probability it is either comprised between 0 and 1 or it is a percentage. In the latter case you need to explicit the % sign.

Answer: Thanks for your advice. We have corrected it.

Author Response

Dear Editors and Reviewers:

Thank you for your comments on the paper. Efforts have been made to clarify the following points and correct the mistakes. The modifications based on the suggestions are explained in further detail below.

1.(Line 15): Atp8 gene is not absent in this species and I’m pretty sure it is not absent in the whole Mytilidae. The sequences for atp8 in Mytilidae are very divergent but it was proven on Mytilus spp. that ATP8 protein is indeed codded by mtDNA.

Zhao B, Gao S, Zhao M, Lv H, Song J, Wang H, Zeng Q, Liu J. 2022 Mitochondrial genomic analyses provide new insights into the “missing” atp8 and adaptive evolution of Mytilidae. BMC Genomics 23, 738. (doi:10.1186/s12864-022-08940-8)

Lubośny M, Przyłucka A, Śmietanka B, Breton S, Burzyński A. 2018 Actively transcribed and expressed atp8 gene in Mytilus edulis mussels. PeerJ 6, e4897. (doi:10.7717/peerj.4897)

When you look at the ATP8 protein sequences in Brachidontinae you can see there is a level of homology between those sequences even though they are often located between different gene sets. I work with bivalve mitogenomes on daily bases and I pay a lot of attention to the correctness of annotations.

Answer: Thanks for your advice. We have corrected it.

2.(Annotations): The length of the annotation of 16S rRNA hasn’t been corrected to the GenBank record ON193524. Also please add the missing atp8 gene.

Answer: Thanks for your advice. We have corrected it.

3.(line 17): “gene order of Mytilisepta is conservative at the genus level”

Answer: Thanks for your advice. We have corrected it.

4.(line 17-18): “However, there is/was/exists a high level of rearrangements in M. virgata compared…”

Answer: Thanks for your advice. We have corrected it.

5.(line 19): “based on concatenated 12 PCGs from Mytilidae”

Answer: Thanks for your advice. We have corrected it.

6.(Line 19-20): “As a result, we found that M. virgata is in the same clade as Mytilisepta spp.” Or ““As a result, we found that M. virgata clusters together with other Mytilisepta spp.

Answer: Thanks for your advice. We have corrected it.

7.(Line 20-21): “it exhibited abundant amount/level of gene rearrangement events at the genus level” or something else…

Answer: Thanks for your advice. We have corrected it.

8.“Estimated times revealed…” “times” of what?? Word is missing here.

Answer: Thanks for your advice. We have changed to “The result of estimated times revealed that the M. virgata and Mytilisepta keenae diverged around the early Paleogene”

9.(line 22): Maybe better word than “but” connecting those two sentences can be used here (although?).

Answer: Thanks for your advice. We have corrected it.

10.(line 24-25): “The findings not only refute previous results…” What “previous results”? Those about Eocene?

Answer: Thanks for your advice. We have modified the sentence.

11.English needs to be corrected. It is very hard to review this in this form. I’m not planning to spend days trying to correct it. I don’t have time for that.

Answer: Thanks for your advice. We have improved our language.

12.(line 30-31): “The mitochondrial (mt) genome of Metazoa is regarded as a good model in the investigation of the phylogenetic of species [or evolution] due to its small molecular weight and maternal inheritance.”

Answer: Thanks for your advice. We have modified the sentence.

13.(line 34): “trendy tool” sounds unscientific maybe “popular”.

Answer: Thanks for your advice. We have modified the sentence.

14.(lines 53-58): Its chaos. Should be split to few separate sentences or rewritten.

Answer: Thanks for your advice. We have modified the sentence.

15.(lines 61-63): “existed” DUI still exists. Also, this sentence sounds bad un relation to sentences written earlier. Maybe something line “Also, an unique system of doubly uniparental mitochondrial inheritance has been discovered in many bivalve species.”

Answer: Thanks for your advice. We have rewritten the sentence.

16.(line 64): “..which was in 1951 by Habe”. Which was what?? Described??

Answer: Thanks for your advice. We have rewritten this sentence.

17.“it/they/this species/those animals usually form a large mussel beds….”

Answer: Thanks for your advice. We have corrected it.

18.(line 70): “this mussel species is flatter ventrally, has wider and…”

Answer: Thanks for your advice. We have corrected it.

19.Please correct English before third revision. I cannot properly review this without English correction.

Answer: Thanks for your advice. We have revised it.

20.(lines 79-80): Not true. Genbank MK721548

Answer: Thanks for your advice. We have corrected it.

21.(line 97) in “N 30°43′1.64″, E 1 97 122°46′3.25″ I don’t know what the “1” stands for.

Answer: Thanks for your advice. We have deleted it.

  1. (line 100) DNA quality is “checked” not “detected”

Answer: Thanks for your advice. We have corrected it.

  1. (lines 119-134): I don’t understand. Put it into full proper sentences.

Answer: Thanks for your advice. We have rewritten it.

24.(line 124) cox1 (italicized)

Answer: Thanks for your advice. We have corrected it.

25.(line 136) “features”. Also, please change the word order in the sentence. The verb should be earlier, and word “revealed” does not fit here.

Answer: Thanks for your advice. We have changed it.

26.(line 146); “quickly” should be deleted.

Answer: Thanks for your advice. We have deleted it.

27.(line 147) “aligned aligned” same word twice

Answer: Thanks for your advice. We have deleted it.

28.(line 152-154): Maybe “The phylogenetic relationships were analyzed by ML method and constructed in IQ-TREE2 using the best-fit GTR+F+R7 model with 1000 nonparametric bootstrapping replicates. The best ML Model was selected based on ModelFinder software results.”

Answer: Thanks for your advice. We have corrected it.

29.(line 184-186): Sentences need to be rewritten.

Answer: Thanks for your advice. We have rewritten it.

30.(Line 187) Atp8 is not missing but it is hard to use this gene in phylogenetic reconstruction in Mytilidae due to the high divergence of this sequence.

Answer: Thanks for your advice. We have re-annotated this sequence.

31.(lines 190-191); calibration points? Or fossils for calibration?

Answer: Thanks for your advice. We have corrected it.

  1. (line 193): remove double spaces

Answer: Thanks for your advice. We have deleted it.

33.(Line 196): “Confirmed the convergence of the chains using Tracer v.1.6” something is missing in this sentence.

Answer: Thanks for your advice. We have rewritten it.

34.(Line 193-199): Was the BEAST run only once? Maybe it would be better to run BEAST like MrBayes in few replicates to make sure results are correct.

Answer: Thanks for your advice. We have tried many times and also the BEAST was run many times before we get this result.

35.(lines 203) “with 33.22% clean data GC after data filtering” with 33.22% GC nucleotide content after filtering?

Answer: Thanks for your advice. We have rewritten it.

36.(line 206) 13 PCGs (atp8)

Answer: Thanks for your advice. We have corrected it.

37.(lines 209-210): “…the reason for all 36 mitochondrial genes being on the heavy strand is unclear.” Is this necessary? Should there be a reason? If there is, I would like to know more about why sometimes all genes are on one strand and sometimes there is a mix.

Answer: Thanks for your advice. We have corrected it.

38.(line 211) GC content is also in line 205.

Answer: Thanks for your advice. We have corrected it.

39.(line 214-215): “The PCG gene had a higher A+T content than the rRNA and tRNA genes among the coding genes.” Which gene? Nad4? What about Nad3?

Answer: Thanks for your advice. We have rewritten it.

40.(lines 233): “Figure 1. Maps of the mitochondrial genomes of M. virgata.” One Map of one genome. I see only one map.

Answer: Thanks for your advice. We have corrected it.

41.(lines 236-239): Grammar.

Answer: Thanks for your advice. We have rewritten it.

42.(Table 1) Genes should be italicized 42

Answer: Thanks for your advice. We have corrected it.

  1. (line 286): “…tRNAs length varied…”

Answer: Thanks for your advice. We have corrected it.

44.(line 294) “existed” ---> “were/are located”

Answer: Thanks for your advice. We have corrected it.

45.(Fig 4) This tRNA-E looks strange only 2 base pairs in anticodon stem.

Answer: Thanks for your advice. We have re-identified tRNA-E and corrected it.

46.(line 329+) lacked annotation for atp8 gene.

Answer: Thanks for your advice. We have corrected it.

47.(lines 348-353): Grammar

Answer: Thanks for your advice. We have rewritten it.

48.(line 354-359) You have used Perna perna sequence (NC_026288) which is indeed a sequence for Mytilaster solisianus (As I mentioned in the first review). This analysis should

be repeated using correct sequences for Perna perna mitogenome. GenBank accession numbers OK576479 (Brazilian specimen B2); OK576480 (western South African specimen) and OK576481 (eastern South African specimen).

Cunha RL, Nicastro KR, Zardi GI, Madeira C, McQuaid CD, Cox CJ, Castilho R. 2022 Comparative mitogenomic analyses and gene rearrangements reject the alleged polyphyly of a bivalve genus. PeerJ 10, e13953. (doi:10.7717/peerj.13953)

Answer: Thanks for your advice. We have rebuilt the tree use MT588202.

47 (lines 360-366): Atp8 is present in all Mytilus species. It is just not annotated in some of them.

Lubośny M, Przyłucka A, Śmietanka B, Breton S, Burzyński A. 2018 Actively transcribed and expressed atp8 gene in Mytilus edulis mussels. PeerJ 6, e4897. (doi:10.7717/peerj.4897)

Answer: Thanks for your advice. We have rewritten it.

  1. You need to rewrite this section (3.4. Gene arrangement ) and do some additional comparisons including presence of atp8 gene in all those species. Including M. keenae. Sequence is located behind atp6 and tRNA-ser. It add additional rearrangement between Mytilisepta spp.

>atp8 M. keenae

ATGAGATTGTTTGGATGTTATAGGTCGGTAGATATTTTTCTCTGAGTGGGTTTTAT TTTTATTATTGTGGAGTTATGTATTTGGTGGGTAGTTCCGGTACGTTTAAAGATGA AATCGTTTATCGTTAAAAGTTTTAGTGGAGTTAGCAAGTTTCCTAATATTTAA

Answer: Thanks for your advice. We have rewritten it.

49.(lines 376-379): Grammar

Answer: Thanks for your advice. We have rewritten it.

50.(line 537): It is only lack of annotation of atp8. If you want to do your analysis properly you should try to identify atp8 gene in all Mytilidae sequences.

51.(line 540) <i>M. virgata</i>

Answer: Thanks for your advice. We have corrected it.

52.(line 541): “Gene segments are drawn to scale”. For me it looks like there is no scale at all. Its only schematic representation of gene order.

Answer: Thanks for your advice. We have corrected it.

53.(lines 548-551): Please rewrite this sentence.

Answer: Thanks for your advice. We have rewritten it.

54.(lines 558): delete “that”.

Answer: Thanks for your advice. We have deleted it.

55.(line 559) “underneath”?

Answer: Thanks for your advice. We have corrected it.

56.(line 560) It is not Perna perna the GenBank record is wrong. It is mtDNA sequence for

Mytilaster solisianus.

Answer: Thanks for your advice. We have corrected it.

57.(lines 576-577): Grammar

Answer: Thanks for your advice. We have rewritten it.

58.(lines 645-647): What? Grammar

Answer: Thanks for your advice. We have rewritten it.

59.(lines 647-649) “The variation in the size of mitochondrial genomes were primarily due to the frequency of duplicated repeats, horizontal gene transfer, genetic drift, and plasmid- derived regions” You didn’t show any data supporting this conclusion, analysing differences between M. virgata and M. keenae. What “duplicated repeats” it is first time it is mentioned?

Horizontal gene transfer?? Plasmid-derived regions where? In Mytilisepta? Show me where are those sequences.

Answer: Thanks for your advice. Our expression is wrong and we have rewritten this sentence.

60.(line 658): regularly used to?

Answer: Thanks for your advice. We have corrected it.

61.(Lines 661): “…base composition ratios were G biased to C…” it sounds bad. Maybe “…were biased towards G…” or change it completely. There were more G than C nucleotides.

Answer: Thanks for your advice. We have rewritten it.

62.(lines 681): there should be a sentence in disunion before/in_front this one mentioning that cox1 and atp6 have truncated stop codon. The answer here is simple tRNA in a way forces the transcript to end without complete stop codon. tRNA creates secondary structures that terminates translation or is just cut out from transcript before protein genes are translated to protein (probably the second part is closer to the truth).

Answer: Thanks for your advice. We have rewritten it.

63.(lines 681-687) this paragraph is pure chaos.

Answer: Thanks for your advice. We have rewritten it.

64.(line 692): maybe other word than “otherwise”

Answer: Thanks for your advice. We have rewritten it.

65 (line 709-711): Grammar

Answer: Thanks for your advice. We have rewritten it.

66.(line713-717): Much is wrong here.

Zhao B, Gao S, Zhao M, Lv H, Song J, Wang H, Zeng Q, Liu J. 2022 Mitochondrial genomic analyses provide new insights into the “missing” atp8 and adaptive evolution of Mytilidae. BMC Genomics 23, 738. (doi:10.1186/s12864-022-08940-8)

Lubośny M, Przyłucka A, Śmietanka B, Breton S, Burzyński A. 2018 Actively transcribed and expressed atp8 gene in Mytilus edulis mussels. PeerJ 6, e4897. (doi:10.7717/peerj.4897)

Answer: Thanks for your advice. We have rewritten it.

67.(lines 773-775): Uliano-Silva made mistake and published sequences for wrong species.

Answer: Thanks for your advice. We have corrected it.

Round 3

Reviewer 4 Report

see attachment

Author Response

Dear Editors and Reviewers:

Thank you for your comments on the paper. Efforts have been made to clarify the following points and correct the mistakes. The modifications based on the suggestions are explained in further detail below.

R3: The complete mitochondrial genome ofMytilisepta virgata(Mollusca: Bivalvia), novel gene rearrangements, and the phylogenetic relationships of Mytilidae

(Line 15): Thirteen (13) protein-coding genes

Answer: Thanks for your advice. We have corrected it.

(Line 16): 13 PCGs

Answer: Thanks for your advice. We have corrected it.

(Line 17): fairly conservative at the genus level. The Atp8 in M. keenae is in different place.

Answer: Thanks for your advice. We have corrected it.

(Line 20): “… in the same clade as other Mytilisepta spp.” OR “…as Mytilisepta keenae”. There is only one other Mytilisepta species available.

Answer: Thanks for your advice. We have corrected it.

(Lines 17-18 vs 20-21): “Compared to the putative… etc.” Sentences are very similar one is about M. viragata the second about Mytilisepta species. It feels a little redundant but you can convince me it is not.

Answer: Thanks for your advice. We have deleted it.

(Line 34): “…have become a popular of phylogenetic in recent years.” You can ask MDPI for a refund this is still not a proper English sentence.

Answer: Thanks for your advice. We have changed it to “there has been a significant reduction in the cost of DNA sequencing, and analyses of whole mitochondrial genomes (mitogenomes) have gained popularity in recent years for phylogenetic investigations”.

(Lines 75-80): I feel there is something wrong with this sentence. Perna spp. got “more attention” because XYZ, Xenostrobus and Limnoperna become invasive, Modiolarca and Mytilisepta “gained less attention”. For me these are not connected. There should be something like Perna is well researched because it is widely used in XYZ, Xenostobus and Limnoperna got scientific attention due to their invasiveness/invasion in the Northern hemisphere but there are not many research articles concerning Mytilisepta etc.

Answer: Thanks for your advice. We have corrected it.

(Line 82): exists… still exists OR “was described in many bivalve species”.

Answer: Thanks for your advice. We have corrected it.

(line 87): maybe delete “in”.

Answer: Thanks for your advice. We have deleted it.

(line 89): “…has wider and firmer shell…..”

Answer: Thanks for your advice. We have corrected it.

(line 92): “a species” add space between a and species

Answer: Thanks for your advice. We have corrected it.

(line 94); maybe “were moved…”

Answer: Thanks for your advice. We have corrected it.

(line 98): “Until now, only linear/non-circular mitochondrial sequences and individual genes were available for…” or something like that.

Answer: Thanks for your advice. We have corrected it.

(line 159-160): Covaris M220 is an equipment not a method. Maybe it would be better to write something like. “… using a physical ultrasonic method (Covaris M220).

Answer: Thanks for your advice. We have corrected it.

(Line 160-162): These lines are not 100% clear to me. I’m not familiar with “leveling”. It might be correct word but I don’t understand it. I guess you meant “complete the leveling of 3’ end with A nucleotide before ligation of the index connectors”? Oooooohhhh…. Maybe it should be “leaving”.…leaving the 3’ adenosine overhang and ligating it with index connectors? Also, I would end sentence after connectors (dot). “The DNA fragments after ligation were amplified by PCR (8cycles)” can be a standalone sentence.

Answer: Thanks for your advice. We have rewritten it.

(lines 162-164): verb is missing. “… was performed?” Maybe split to two separate sentences?

Answer: Thanks for your advice. We have rewritten it.

(line 164-165): verb is missing

Answer: Thanks for your advice. We have rewritten it.

(Line 166): “…quality value of sequencing reads…”?

Answer: Thanks for your advice. We have rewritten it.

(Lines 160 – 176): Please check the whole 2.2 section. This is still hard to understand. Language corrections needed.

Answer: Thanks for your advice. We have rewritten it.

(Line 181-182): Sentence should be rearranged. Maybe “The sequence features of the mitochondrial circular genome of M. virgata were shown using online CGView Server.”

Answer: Thanks for your advice. We have rewritten it.

(line 183-185): In my opinion this sentence is not clear.

Answer: Thanks for your advice. We have rewritten it.

(line 195): Maybe you should add somewhere here sentence like this “Due to the high divergence the atp8 gene was excluded from this analysis.”

Answer: Thanks for your advice. We have rewritten it.

(line 288-292): double space. Small language corrections needed here.

Answer: Thanks for your advice. We have rewritten it.

(Line 413): “The raw sequencing data…” please change order of the words

Answer: Thanks for your advice. We have rewritten it.

(line 413-414): I think number of sequencing reads would be more informative than the size of a .fastq data file.

Answer: Thanks for your advice. We have added it.

(Line 678 and Table 1): There is discrepancy between AT in table vs the values in the text. Please check that. If I interpret it correctly, AT-skew is negative and GC-skew is positive.

Answer: Thanks for your advice. We have corrected it.

(line 685): I not sure about this statement: "...the two rRNA genes (rrnL and rrnS) were located between trnF and trnQ in the M. virgata mitogenome". This is true but is this information important? Nd4l, nd3 and few tRNAs are also between trnF and trnQ. You can leave it as it is but it’s not very informative to me. Maybe it would be better to write what is flanking rrnL and rrnS separately? The choice is yours.

Answer: Thanks for your advice. We have rewritten it.

(line 754): Did you mean “Mytilinae”?

Answer: Thanks for your advice. We have corrected it.

(Line 755): “mitotic genome”? like mitosis? Please check if this word is correct in this context.

Answer: Thanks for your advice. We have cerrected it.

(Line 777): "The gene arrangement in the mitogenome of M. virgata was found to be identical to that of M. keenae,". Unfortunately, this is not true and if I remember correctly, I pointed this out to you in one of the earlier reviews. In M. keenae there is cox2-atp6-atp8 but in your mitogenome there is cox2-atp8- atp6. Additional corrections or disclaimers to figures 5 and 6 (There is no atp8 box in the gene order for M. keenae) will be needed.

Answer: Thanks for your advice. We have corrected it.

(line 866 – 867): “green”? maybe the picture quality is bad but for me it looks orange.

Answer: Thanks for your advice. We have corrected it.

(line 879-881): I don’t understand. Please make it clearer. Maybe add verbs, rearrange the sentence or divide it in into few shorter ones.

Answer: Thanks for your advice. We have rewritten it.

(line 910-911): “Phylogenetic tree inferred using Bayesian inference (BI) and maximum likelihood (ML) methods, the PP value is in front of the node. The value on the left side of the slash is the posterior probabilities estimated by the Bayesian tree, and the value on the right side is the maximum likelihood tree.” I think the bolded part can be removed.

Answer: Thanks for your advice. We have deleted it.

(Fig 8): “Calibration taxa are indicated with an asterisk on the corresponding nodes, and the asterisk indicates M. virgatein this study, respectively.” You use asterisk (star sign) for two different things it is confusing. Please change one of the symbols or write something about colors. I would prefer for you to change symbol because of potential colorblind researchers. You have used two calibrations points. Am I wrong thinking there should be an additional asterisk for the second calibration point? One around 78 Mya and second around 334 Mya?

Answer: Thanks for your advice. We have corrected it.

(Line 1076): “…bivalve Mollusca”?? What is the intention here? You want to point that bivalves are molluscs (“…bivalves (Mollusca)”)? Or is GC positive skew common in most known molluscs and especially bivalves? It is not clear here, please change this sentence a little add “(…)” or something.

Answer: Thanks for your advice. We have rewritten it.

(Lines 1081-1083): The fact that “this phenomenon has been observed in other metazoan mitogenomes “is not an explanation for codon usage bias. I think this should be a separate sentence.

Answer: Thanks for your advice. We have rewritten it.

(Lines 1209 – 1210): I don’t see the connection here. “tRNAs were highly conserved in the nuclear genome…”. The usually are but you have not analyzed nuclear tRNA for this species. This sentence is not well connected to the previous one and the change from present tense to past tense made me think that you are writing about nuclear tRNA for M. virgata. Please check this paragraph carefully.

Answer: Thanks for your advice. We have rewritten it.

(Line 1232) “the short” -> “too short”.

Answer: Thanks for your advice. We have corrected it.

(Line 1238):

  1. Citation [74] is not “Marek et al.”

“[74] Marín, A., Fujimoto, T., Arai, K. The mitochondrial genomes of Pecten albicans and Pecten maximus (Bivalvia: Pectinidae) reveal 1791 a novel gene arrangement with low genetic differentiation. Biochem Syst Ecol. 2015, 61, 208-217. https://doi.org/10.1016/j.bse.2015.06.015”

  1. It is not Marek et al. it is Lubośny et al. Marek is the name of the author and Lubosny is the

surname. I guessing you meant the article (bellow) I have pointed you toward in the last review.

Lubośny M, Przyłucka A, Śmietanka B, Breton S, Burzyński A. 2018 Actively transcribed and expressed atp8gene in Mytilus edulismussels. PeerJ6, e4897. (doi:10.7717/peerj.4897)

  1. Also this paper is not in your References. There is no Marek et al. article in your references.

Please check ALL your references very carefully again. They might be pointing to wrong

articles

Answer: Thanks for your advice. We have corrected it.

(Line 1383): The same thing with “Combosch et al.” and [84]. They are not the same. Combosch is [81].

Answer: Thanks for your advice. We have corrected it.

(line 1385). Combosch used only sequences for M. virgata and B. exustus. There were no sequences for P. purpuratus and S. algosus/patagonicus so there is no real inconsistency.

Answer: Thanks for your advice. We have corrected it.

(Line 1557): looking at Fig.4 DHU stem is present in the trnS1 and you don’t talk about trnS1 anywhere in the text. Is this sentence correct?

Answer: Thanks for your advice. We have corrected it.
